# SolarCube: An Integrative Benchmark Dataset Harnessing Satellite and In-situ Observations for Large-scale Solar Energy Forecasting

**Ruohan Li**[1], **Yiqun Xie**[1]*, **Xiaowei Jia**[2], **Dongdong Wang**[1], **Yanhua Li**[3], **Yingxue Zhang**[4], **Zhihao Wang**[1], **Zhili Li**[1]

[1]University of Maryland, [2]University of Pittsburgh, [3]Worcester Polytechnic Institute
[4]SUNY - Binghamton
{r526li, xie, ddwang, zhwang1, lizhili}@umd.edu,
xiaowei@pitt.edu, yli15@wpi.edu, yzhang42@binghamton.edu

## Abstract

Solar power is a critical source of renewable energy, offering significant potential to lower greenhouse gas emissions and mitigate climate change. However, the cloud induced-variability of solar radiation reaching the earth's surface presents a challenge for integrating solar power into the grid (e.g., storage and backup management). The new generation of geostationary satellites such as GOES-16 has become an important data source for large-scale and high temporal frequency solar radiation forecasting. However, no machine-learning-ready dataset has integrated geostationary satellite data with fine-grained solar radiation information to support forecasting model development and benchmarking with consistent metrics. We present SolarCube, a new ML-ready benchmark dataset for solar radiation forecasting. SolarCube covers 19 study areas distributed over multiple continents: North America, South America, Asia, and Oceania. The dataset supports short (i.e., 30 minutes to 6 hours) and long-term (i.e., day-ahead or longer) solar radiation forecasting at both point-level (i.e., specific locations of monitoring stations) and area-level, by processing and integrating data from multiple sources, including geostationary satellite images, physics-derived solar radiation, and ground station observations from different monitoring networks over the globe. We also evaluated a set of forecasting models for point- and image-based time-series data to develop performance benchmarks under different testing scenarios. The dataset is available at `https://doi.org/10.5281/zenodo.11498739`. A Python library is available to conveniently generate different variations of the dataset based on user needs, along with baseline models at `https://github.com/Ruohan-Li/SolarCube`.

## 1  Introduction

Solar power has emerged as a critical source of renewable energy, offering significant potential to reduce the impact of climate change by lowering greenhouse gas emissions. Over the past decade, solar power has steadily grown, now accounting for 27% of renewable energy generation [1]. It is projected by the International Energy Agency's Sustainable Development Scenario that 4,240 GW of photovoltaic solar generating capacity will be deployed by 2040 [2]. However, the variability of the amount of solar radiation that reaches the earth's surface presents a challenge for integrating solar power into the grid [3]. Limited forecasting skills for the fluctuation increase the costs due to the need for larger backup battery sizes capacities [4] and the operation of peaking plants [5].

---

*Corresponding author.

38th Conference on Neural Information Processing Systems (NeurIPS 2024) Track on Datasets and Benchmarks.

Such fluctuation will also influence power generation, transmission, and distribution scheduling [6]. Accurate forecasts of solar radiation are therefore essential to help grid operators manage this variability more effectively, promoting more efficient and reliable use of solar energy [7].

Based on the needs of solar energy applications and available data sources, current tasks in solar forecasting can be classified from both temporal and spatial perspectives [8]. In terms of forecasting horizons, they can be categorized into ultra-short-term (seconds to 30 minutes) for real-time energy dispatch [6], short-term (30 minutes to 6 hours) for integrated power systems operation and management [9], and long-term (day ahead and beyond) for power generation, transmission, and distribution scheduling [10]. In terms of forecasting spatial scales, most of the existing studies focus on point-based forecasting, which is beneficial for specific photovoltaic utility planning using measurements from ground monitoring stations at certain locations. At the same time, there is an increasing need for area-based forecasting for large-scale photovoltaic system management and energy transportation [7].

This dataset focuses on forecasting total shortwave radiation reaching the Earth's horizontal surface, which will be termed solar radiation for simplicity. The fluctuation of solar radiation is influenced by both the predictable diurnal and seasonal cycles of the solar zenith angle (SZA) and the challenging-to-forecast movement of clouds. It is reported that a disruption in dense and fragmented cloud cover can lead to a significant surge, up to 700 $W/m^2$ within a 30-min timeframe [11]. Hence, studies started to incorporate spatial information for higher forecasting capacity. For ultra-short-term forecasting, sky images that take photos of the sun from the ground at second intervals fulfill the need [12]. To forecast a longer time horizon, the new generation of geostationary satellites have become an important data source [9]. Unlike polar-orbiting satellites, geostationary satellites offer a much higher temporal resolution, providing data at intervals as short as 5 minutes. This allows for monitoring of cloud movements from distant areas and enables forecasting from 15 minutes to several days in advance [11]. In addition, geostationary satellites provide multi-continental coverage, which is important for managing extensive photovoltaic systems and energy transportation beyond the limited regions with ground monitoring stations.

In related work, a sky-image-based dataset for solar radiation forecasting was developed [12], but it is specifically designed for very short-term (e.g., seconds to 30 min) tasks. Boussif et al. [13] published the data used to develop a multi-modal approach for solar radiation forecasting at six stations in Europe, including satellite images that provide cloud information around the stations. However, it has a coarse spatial resolution at ∼0.5 degrees (50km), which is limited for capturing dynamic changes driven by fragmented clouds for short-term solar radiation forecasting. Moreover, the dataset is still designed for point-based forecasting. While reanalysis datasets based on physical simulations, such as ECMWF Reanalysis v5 (ERA5), have been widely used in weather forecasting tasks [14], existing studies have shown that their solar outputs are less accurate compared to those produced primarily based on satellite observations and radiation-focused physics models [15]. The hourly updates of existing reanalysis datasets also do not meet the requirements for short-term forecasting. Other related ML-ready datasets, such as SEVIR [16], have also incorporated geostationary satellite observations and physics-derived products to enable area-based forecasting. However, they primarily focus on precipitation and are organized around storm events, which are not associated with high solar radiation conditions.

To address the above-mentioned issues, we present SolarCube, a new ML-ready benchmark dataset for renewable energy applications. The dataset is composed of geostationary satellite images, a solar radiation product based on the satellite images, and measurements from ground monitoring networks. SolarCube covers 19 study areas distributed over multiple continents: North America, South America, Asia, and Oceania. The selection of the study areas is based on several factors, including the spatial coverage of the existing geostationary satellites considered (e.g., GOES-16), the spatial distribution of the ground monitoring sites, the diversity of landscapes (e.g., croplands, urban area), etc. Each study area covers a 600 km × 600 km region, and carries year-long sequences of solar radiation data at a 15-minute temporal resolution (i.e., over 35,000 time steps), as shown in Fig. 1, to fulfill the need of the diverse forecasting tasks. The main contributions of our work are summarized below:

- SolarCube provides ML-ready geostationary satellite observations, satellite-based solar radiation products, and ground measurements for both point- and area-based short- and long-term solar radiation forecasting.

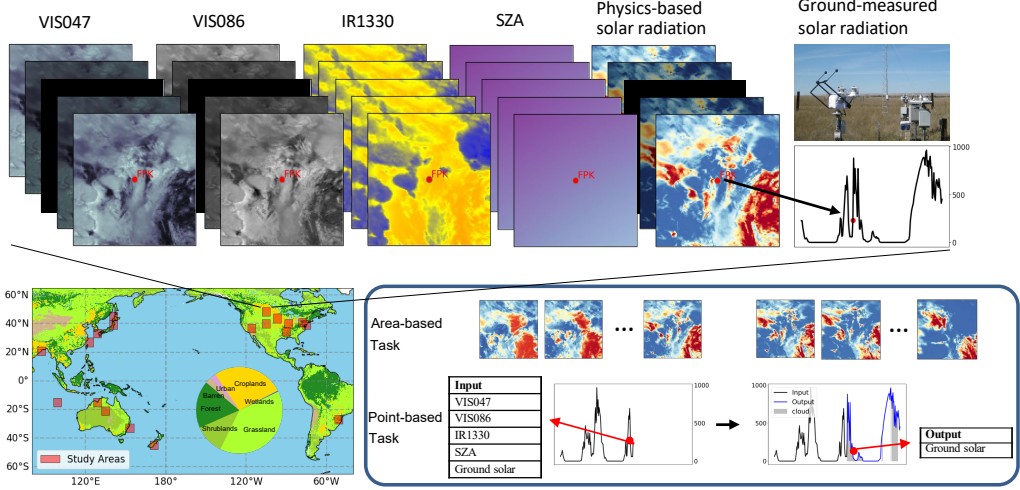

Figure 1: SolarCube dataset composition, study areas, and baseline tasks.

- We propose consistent metrics to evaluate a set of forecasting models for point-based and area-based time-series data, creating benchmarks for forecasting quality.

- We develop additional geo- and physical-context data including land cover and cloud masks as part of SolarCube for scenario-based testing.

- We provide a Python package for SolarCube to allow users to customize the input and output time length, data format, and aggregation level to easily generate data for other variations of the tasks.

## 2  Related Work

**Solar radiation forecasting.**    Deep learning models, such as LSTM and transformers, are applied to solar radiation forecasting due to their capability to model intricate nonlinear patterns and capture temporal dependencies. Traditional methods, including LSTM and CNN, are extensively used for both short-term and long-term forecasting [17; 18; 19]. With recent progress in transformer, a self-attention-based multi-horizon model for solar forecasting was proposed [20]. [13] further integrated cross-video vision transformer to combine satellite images and ground measurements for day-ahead forecasting. Additionally, [21] used Informer to test the optimal combination for point-based solar radiation forecasting. At image scales, [22] developed IrradianceNet using ConvLSTM on satellite-derived solar radiation data for 4h ahead forecasting in Europe. [7] used the predictive recurrent neural network (PredRNN++) with Himawari-8/9 satellite for solar energy forecasting over China.

**Earth System forecasting.**    With the availability of datasets such as ERA5 [23] and SEVIR[16], data-driven methods have been increasingly adopted in Earth system forecasting tasks. [24] developed ConvLSTM for short-term precipitation forecasting, which was further improved by several models such as PredRNN [25], Trajectory GRU [26], etc. [27] further proposed an EarthFormer for earth system forecasting in general, which leverages various building cuboids (e.g., video Swin, global vector) to integrate local and global information. 3D Earth-specific transformer (3DEST) [14] and FourCastNet [28] have also been developed for short-to-medium range global weather forecasting.

**Existing datasets**    For solar radiation forecasting, ground-based sky images have been utilized for ultra-short-term forecasting (seconds to 30 min) [29; 30]. However, these sky images are limited to specific locations where the sensors are installed, typically covering a radius of around 5-10 kilometers. Observations from ground monitoring stations with additional image-modal inputs as auxiliary features have been used for long-term solar radiation forecasting [13]. Similarly, forecasting from the ground-based dataset is limited to 6 stations in Europe, and does not support tasks at larger geographic extents with more complete spatial coverage. The ML-ready precipitation dataset, SEVIR [16], has incorporated geostationary satellite observations, enabling area-based forecasting. However,

it was designed specifically for precipitation forecasting and is organized around storm events, which are not associated with high solar radiation conditions. Additionally, the dataset does not include data from ground measurements and is therefore not applicable to solar radiation forecasting.

**Limitations of existing work.** While existing work has shown ML's potential for solar radiation forecasting, the current datasets are often small in scale with limited geographic coverage (e.g., only at a few points/locations within a geographic region), and designed for a specific solar forecasting task. On the other hand, datasets covering large geographic areas are mostly centered around other problems (e.g., precipitation) that are distinct from solar radiation forecasting (storm vs. non-storm events). SolarCube aims to address the gaps by developing an integrative, ML-ready benchmark dataset to facilitate the development of solar radiation forecasting models. The dataset will support both area-based forecasting using large-footprint satellite imagery and derived solar data, as well as point-based forecasting with ground measurements from monitoring stations at various lead times.

## 3  Dataset Construction

Fig. 1 shows an overview of SolarCube and the forecasting tasks. The main data include images from geostationary satellites, physics-based satellite solar radiation, and ground measurements from networks of monitoring stations. Additional data include solar zenith angle and cloud mask at each time step, the land cover types, and the geo-coordinates of the study areas.

**Geostationary satellite data collection.** We extracted and combined three spectral bands at the wavelengths of 0.47 μm (VIS047), 0.86 μm (VIS086), and 13.3 μm (IR1330) from two different geostationary satellites: GOES-16 and Himawari-8. The choice of the three bands is based on the previous study showing that they have the highest correlation with the surface solar radiation target [31; 32]. Visible bands (i.e., VIS047 and VIS086) are only available in the daytime. The infrared band (i.e., IR1330) is continuous all day long, which provides useful information for day-ahead forecasting. We downloaded the images from the National Aeronautics and Space Administration (NASA) GeoNEX portal, where the original geostationary data have been processed with radiometric calibration and geometry calibration [33]. We selected 19 study areas, each of size 600 km × 600 km, around the ground monitoring stations to download the images. Next, we process the original images with a spatial resolution of 1 km by aggregating them into a 5 km resolution. This is supported by previous studies that showed solar radiation has high homogeneity within 10 km and the aggregation improves the prediction quality by considering the spatial autocorrelation [34; 35]. Moreover, the aggregation also substantially reduces the data size, making it easier to use. Finally, since the GOES-16 data has a 15-minute temporal resolution while the Himawari-8 data has a 10-minute temporal resolution on the GeoNEX portal, we averaged the Himawari-8 data around the 15-minute mark to ensure data from both sensors have a consistent temporal resolution.

**Physics-based solar radiation data generation.** Retrieving surface radiation from satellite images has long been a focus of the remote sensing communities [36][37][31], leading to products with significantly better accuracy and resolution than reanalysis data [38; 15]. In SolarCube, we generate the first satellite-derived solar radiation data at a 15-min resolution interval, based on two radiative transfer parameterization schemes [39]. The first scheme establishes a relationship between the top of atmosphere reflectance received by the satellite and atmospheric conditions. The second scheme parameterizes atmospheric conditions with the surface-received radiation. The simulated results from the radiative transfer models are stored in two Look-Up Tables (LUTs), which allow for the estimation of surface radiation by matching the spectral band values from the satellites. In addition to the spectral bands from the geostationary satellites, we also prepared several other necessary inputs including surface reflectance from MODIS/Terra+Aqua Albedo (MCD43) [40] and albedo climatology data [41], elevation from [42], and total precipitable water vapor from MERRA2 [43]. More details on the methods and input variables can be found in [31; 35]. Our previous work has successfully used this algorithm to generate the NASA MODIS/Terra+Aqua Surface Radiation product (MCD18) at daily/3-hour resolution [44] and the GeoNEX DSR/PAR Surface Radiation product at hourly resolution [35]. The accuracy and generality of these products have been comprehensively validated, and the results outperformed other existing radiation products [35]. For the first time, SolarCube utilized this mature algorithm to generate radiation data at a 15-minute temporal resolution. To ensure the quality of the physics-based solar radiation data generated in SolarCube, we validated it using

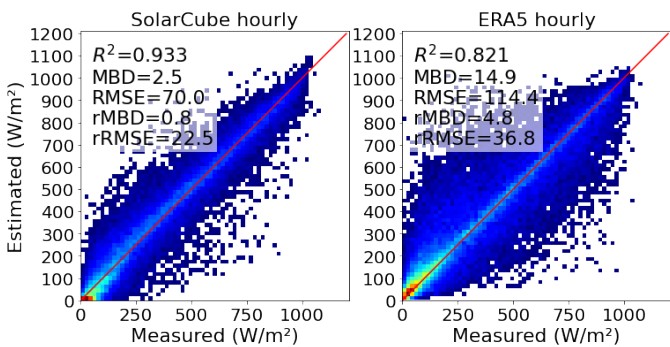

Figure 2: Comparasion of SolarCube and ERA5 at hourly scale. For the definitions of MBD, RMSE, rMBD, and rRMSE, please see Sec. 4.1.

ground measurements from 19 sites and compared it with ERA5, after aggregating the data to an hourly scale for consistency with ERA5. As shown in Fig. 2 , SolarCube demonstrates significantly better performance than ERA5, with the color of the scatterplot representing sample density.

**Insitu data processing.** The ground measurement data were collected from 19 sites across two networks BSRN and SURFRAD, which are within the 19 study areas selected. The data is collected by pyrheliometers and pyranometers, and the reported instrument operating errors are around 2% for pyrheliometers and 5% for pyranometers – or 15 W/m$^2$ [45]. We first followed each network's preprocessing instructions to ensure data consistency and quality. The data passed the BSRN quality check [46] and the procedure described by [34]. BSRN and SURFRAD collect data at 1-minute intervals. To mitigate representative errors between the point-based measurements and satellite image pixels[47], the measured solar radiation was averaged over a 15-minute window centered at the satellite passing time.

**Solar Zenith Angle.** SZA is the angle between the vertical line at a specific location and the line connecting that point to the sun. It indicates how high the sun appears in the sky and decides the maximum possible solar radiation under the clearest sky conditions. SZA varies with time of day, geographic location, and season, which is a critical parameter in calculating solar radiation. We calculated and generated gridded SZA at 15 minute time scale and 5 km resolution, to match it with other data.

**Auxiliary data for scenario-based evaluation.** We additionally include data on clouds and land cover types to construct scenarios with different levels of difficulty for evaluation. For cloud masks, the NOAA GOES-R Series Level 2 clear sky masks and EUMETSAT NWC SAF Himawari-AHI, Cloud Mask data were included to evaluate model performance under two scenarios: (1) Easy: consistent cloud cover, and (2) Hard: dynamic cloud cover. The products provide binary information indicating the presence or absence of clouds at each pixel at specific times. The data were generated with the same temporal resolution as GOES-16 and Himawiri-8. The land cover type data are downloaded from MODIS/Terra+Aqua Land Cover Type (MCD12) [48]. Both datasets provide coverage for all study areas in SolarCube.

**Dataset Splitting.** Solar radiation forecasting patterns can vary across geographic regions due to differences in land types and climate patterns, which influence cloud movement and how radiation is received and reflected at the surface. Therefore, it is essential to perform the train-test split based on distinct geographic regions. We use the data of 14 study areas as training and the rest 5 as testing, which can better indicate the models' generalizability over other regions considering spatial variability [49]. We do not split training and testing data based on the time dimension to ensure sufficient training data to capture the strong diurnal and seasonal cycles of solar radiation.

**Data usage and ML tasks.** The dataset is designed to cover diverse solar forecasting tasks. Based on the spatial coverage and applications, we consider two types of tasks: (1) area-based forecasting for large-scale photovoltaic system management using physics-derived solar data from satellite

images, and (2) point-based forecasting for specific photovoltaic utility planning using measurements from ground monitoring stations at certain locations. These tasks will be detailed in Sec. 4. In terms of the forecasting horizon, using high-quality ground measurements offers better stability for forecasting using longer ranges. Therefore, we consider both short-term and long-term forecasting for point-based forecasting. For area-based forecasting, we follow the settings of current studies, which primarily considered the more feasible short-term ranges [7]. With that said, the data we include in SolarCube offers flexibility for area-based long-term forecasting for interested users. We provide a Python package that allows users to customize the input and output data length, format, and data source, enabling various combinations.

## 4 Experiments

This section presents the evaluation results on two tracks: area-based and point-based forecasting.

### 4.1 Track 1: Area-based forecasting.

**Task formulation.** Let $\mathbf{Y}_t \in \mathbb{R}^{w \times h}$ represent the $t$-th frame in a sequence of physics-derived solar radiation images, where $w$ and $h$ denote the width and height, respectively. As shown in Fig. 3, given the input of the previous $n$ frames $\mathbf{Y}_{t-n+1:t} = (\mathbf{Y}_{t-n+1}, \mathbf{Y}_{t-n+2}, \ldots, \mathbf{Y}_t)$, our goal is to forecast the future $m$ frames $\mathbf{Y}_{t+1:t+m} = (\mathbf{Y}_{t+1}, \mathbf{Y}_{t+2}, \ldots, \mathbf{Y}_{t+m})$. These images only record solar radiation values, which are univariate. In this track, we use the previous 3-hour images to forecast images in the next 3 hours. The sequences are created with the 15-minute time interval, resulting $n = m = 12$. Each image is in the size of 120×120, i.e., $w = h = 120$, representing a $600 \times 600$ km area. The physics-derived solar radiation data from satellite images is present at every pixel in each image. Here we do not utilize other input forcing.

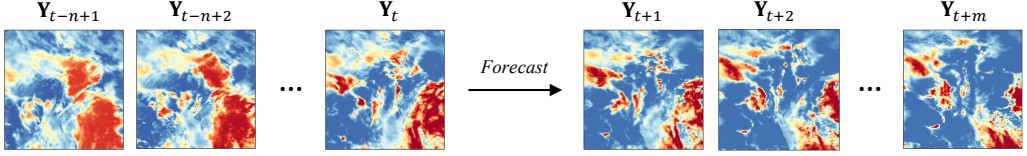

Figure 3: Area-based forecasting task demonstration.

**Models.** The candidate models for this track are ConvLSTM [24] and video transformer models including axial attention (Axial) [50], video swin-Transformer (Video-swin) [51], and divided space-time attention (Divided-st) [52]. The video transformer models are constructed using the structure of Earthformer [27]. The detailed model configurations can be found in the supplementary documents.

We also include the persistence model for comparison, which is a common baseline that forecasts based on temporal autocorrelation [53]. Let $\mathbf{k}_t \in \mathbb{R}^{w \times h}$ denote the derived image of the clearness index, where $t$ is the last time step before forecasting. $\mathbf{k}_t = \mathbf{Y}_t / \mathbf{R}_{\max,t}$ is basically the observed solar radiation $\mathbf{Y}_t$ at time $t$, normalized by the maximum possible solar radiation under clear-sky conditions $\mathbf{R}_{\max,t}$. This maximum radiation only depends on SZA at each location and is calculated as $\mathbf{R}_{\max,t} = E \cdot \cos(\boldsymbol{\theta}_t)$, where $E$ is the solar constant, set to 1360 W/m$^2$ in this context, and $\boldsymbol{\theta}_t \in \mathbb{R}^{w \times h}$ contains the SZA values. For short-term forecasting, at each forecast lead time $t + i$, the short persistence model assumes constant atmospheric conditions represented by a fixed clearness index, $\mathbf{k}_t$, at the last time step $t$ before forecasting. The forecasts are made by varying maximum possible radiation values over different forecasting time steps:

$$\mathbf{Y}_{\text{short\_persistence}} = (\mathbf{k}_t \cdot \mathbf{R}_{\max,t+1}, \mathbf{k}_t \cdot \mathbf{R}_{\max,t+2}, \ldots, \mathbf{k}_t \cdot \mathbf{R}_{\max,t+m})$$

**Evaluation methods and metrics.** Solar radiation follows strong diurnal and seasonal cycles, and its dynamics vary across geographic locations. Therefore, using absolute root mean square error (RMSE) and mean bias deviation (MBD) is not ideal for comparison across different scenarios (e.g., stations, areas, seasons, times, and cloud conditions). The relative value of these metrics, rRMSE and rMBD, normalized to account for variations in solar radiation patterns, provides a

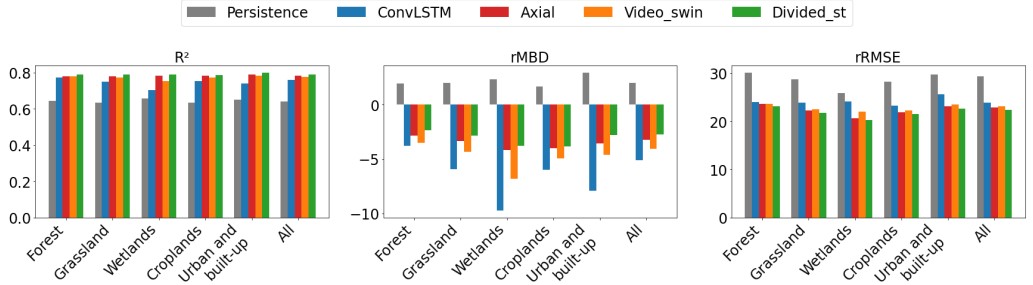

Figure 4: Performance metrics for area-based short-term forecasting under different land cover types.

more robust basis for comparison, with rMBD $= 100 \cdot \text{MBD}/(N^{-1} \cdot \sum_{i=1}^{N} y_i)$, and rRMSE $= 100 \cdot \text{rRMSE}/(N^{-1} \cdot \sum_{i=1}^{N} y_i)$, where $y_i$ is the true value and $N$ is the total number of observations. These relative metrics have been widely used in solar radiation estimation studies [15; 34].

Moreover, it is worth mentioning that the validation results for solar radiation could change significantly when different spatial and temporal scales are considered [34]. Hence, it would be important to highlight the spatial and temporal scale for each validation result. In addition, we consider another widely used evaluation metric, Forecasting Skill ($FS$) [54], to better compare with the persistence model results. This metric is calculated as:

$$FS = (1 - \text{rRMSE}/\text{rRMSE}_{\text{persistance}}) \times 100$$

For short-term forecasting, we use cloud masks as explained in the auxiliary data part in Sec. 3 to separate all testing samples in two scenarios: (1) Easy: The cloud mask at $t + m$ is the same as the cloud mask at $t$; (2) Hard: The cloud mask at $t + m$ is different from the cloud mask at $t$.

**Results.** Table 1 summarizes the performance of different models for area-based forecasting. The deep learning models achieve comparable performance when tested over all samples, with Divided-st showing the highest $R^2$ and lowest rRMSE. In contrast, the persistence model has the lowest absolute rMBD. When validated over Hard scenarios, ConvLSTM achieves the highest $R^2$ and lowest rRMSE. Over easy scenarios, Divided-st shows the lowest rRMSE. Overall, Divided-st has more consistent FS across both hard and easy scenarios.

Fig. 4 compares the model performance under different surface types covered by the testing areas. The results show that all models exhibit similar performance over different surface types in terms of $R^2$ and rRMSE. However, rMBD varied across different land types, with most underestimations occurring over wetlands, corresponding to the limited data sample over these surface types.

Fig. 6 (a) shows the FS value across various forecasting horizons. The ConvLSTM model achieves a high FS at the initial time step, but it declines significantly as the forecasting time increases. Video-swin presents the reverse results, with increased FS as forecasting time increases from 15 minutes to 1.5 hours, and decreased FS as the forecasting time further increases beyond 1.5 hours. Among them, Divided-st shows constantly high FS across all forecasting horizons.

Table 1: Performance metrics for area-based short-term forecasting under various conditions. rMBD, rRMSE, and FS are in percent (%).

| Models | All | | | | Hard | | | | Easy | | | |
|---|---|---|---|---|---|---|---|---|---|---|---|---|
| | $R^2$ | rMBD | rRMSE | FS | $R^2$ | rMBD | rRMSE | FS | $R^2$ | rMBD | rRMSE | FS |
| Persistence | 0.671 | **1.98** | 29.09 | - | 0.392 | **1.37** | 26.91 | - | 0.677 | 2.25 | 29.96 | - |
| ConvLSTM | 0.771 | -5.15 | 23.84 | 18.05 | 0.595 | -6.43 | 21.94 | 18.44 | 0.782 | -4.60 | 24.60 | 17.89 |
| Axial | 0.785 | -3.28 | 22.77 | 21.73 | 0.619 | -6.01 | 21.08 | 21.64 | 0.800 | -2.45 | 23.01 | 23.20 |
| Video-swin | 0.782 | -4.29 | 23.09 | 20.64 | **0.653** | -1.82 | **20.13** | **25.19** | 0.793 | **2.19** | 23.39 | 21.93 |
| Divided-st | **0.792** | -2.81 | **22.32** | **23.28** | 0.646 | -4.55 | 20.32 | 24.49 | **0.805** | -2.42 | **22.69** | **24.25** |

## 4.2 Track 2: Point-based forecasting

**Task formulation.** Let $\mathbf{X}_t \in \mathbb{R}^4$ represent the 4-dimensional feature vector at time $t$, where the features include three spectral bands from geostationary satellites and SZA. Let $\mathbf{Y}_t^p \in \mathbb{R}$ represent

the solar radiation at point-level provided by ground measurements at time $t$. The objective is to predict solar radiation for the next $m$ time steps, $\mathbf{Y}_{t+1:t+m}^p = (\mathbf{Y}_{t+1}^p, \mathbf{Y}_{t+2}^p, \dots, \mathbf{Y}_{t+m}^p)$, using the past $n$ input sequences of features $\mathbf{X}_{t-n+1:t}$ and corresponding measured solar radiation $\mathbf{Y}_{t-n+1:t}^p$, as shown in Fig. 5.

We have two sub-tasks here for short-term and long-term forecasting at the point-level. These sub-tasks have the same formats of inputs and outputs but differs in the sequence length. For short-term forecasting, we set $n = m = 12$, to test the model forecasting for the next 3 hours based on data from the previous 3 hours. For long-term forecasting, we have $n = m = 96$, to test the model forecasting for the next 24 hours based on the data from the previous 24 hours.

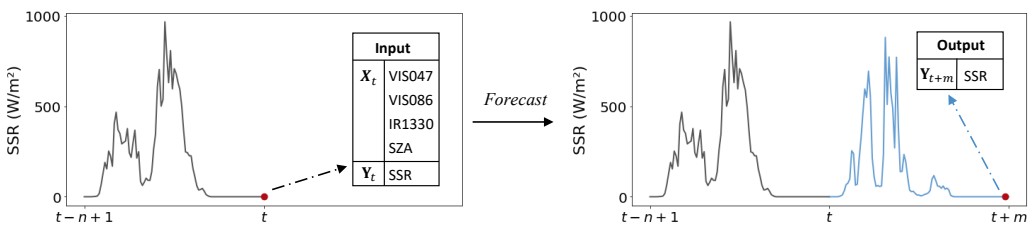

Figure 5: Point-based forecasting task demonstration. SSR represents ground measured solar radiation.

**Models.**   The candidate models we include for this track are LSTM [55], LSTM with time-series attention (LSTM-a) [56], Transformer [57], and Informer [58]. The detailed configuration of the models can be found in the supplementary documents. For short-term forecasting, we adopt the same persistence model defined in Sec. 4.1. The only difference is that the inputs and outputs at each time step is a scalar (point) instead of images (area). Similarly, for day-ahead long-term forecasting, we use a long persistence model as a baseline, which assumes atmospheric conditions remain the same as those at the corresponding time on the previous day. For each forecast lead time $t + i$, we calculate $\mathbf{k}_{t+i-d}^p$ (i.e., clearness index at point-level), where $d = 24 \times 4 = 96$ corresponds to 96 time steps (the same time on the previous day). We then multiply $\mathbf{k}_{t+i-d}^p$ by the forecasted day's maximum possible solar radiation under clear-sky conditions at different time steps:

$$\mathbf{Y}_{\text{long\_persistence}} = (\mathbf{k}_{t+1-d}^p \cdot \mathbf{R}_{\text{max},t+1}^p, \mathbf{k}_{t+2-d}^p \cdot \mathbf{R}_{\text{max},t+2}^p, \dots, \mathbf{k}_{t+m-d}^p \cdot \mathbf{R}_{\text{max},t+m}^p)$$

**Evaluation methods and metrics.**   We used the same evaluation metrics defined in Sec. 4.1. For short-term forecasting, we evaluate the result under the Easy and Hard scenarios defined in Sec. 4.1. For long-term forecasting, the Easy and Hard scenarios are defined following [13]. The Easy represents minimal changes in average solar radiation across consecutive days, while Hard represents significant changes of solar radiation from the previous day.

**Results.**   Table 2 summarizes the point-based short-term forecasting results under different scenarios. Overall, all deep learning models exhibit better rRMSE than the persistence model, demonstrating their enhanced ability to capture cloud-induced solar variation. However, there are no significant improvements of deep learning models from hard to easy cases. Among the models, the Transformer shows the best results for all three metrics across all testing samples, but the performance differences among these four models are not substantial. Fig. 6 (b) presents the FS across the forecasting horizons. All models exhibit a similar trend, with FS increasing as the time step increases, highlighting the improved capability of deep learning models in longer time step forecasting within intra-day periods. The LSTM-a model's FS scores for the initial time steps are more variable compared to the curves of the other models.

For long-term forecasting as shown in Table 3, the persistence model degrades significantly, which is expected as the cloud conditions at the same time on the previous day differ from those on the next day. This issue is particularly pronounced under Hard scenarios. In such situations, deep learning models demonstrate a much stronger ability in solar radiation forecasting. Among the models, LSTM-a shows better performance of all metrics over all samples. This high accuracy largely stems from the

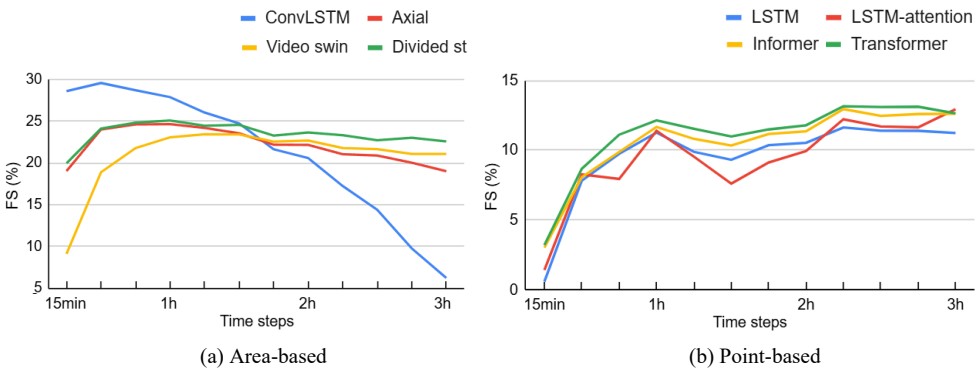

|  | (a) Area-based | (b) Point-based |

Figure 6: Forecasting skill over time steps for short-term area-based and point-based forecasting.

high accuracy at predicting over easy cases. Over hard scenarios, LSTM-a, Informer, and Transformer present similar results on the three metrics. LSTM shows the worst result at long-term forecasting for all cases. We also validate the result on a daily scale to provide the benchmark for application in day-ahead energy planning. After aggregating into the daily scale, Transformer shows the lowest rRMSE and highest $R^2$ with satisfied rMBD. Fig. 7 shows the FS over the 24-hour time steps. The difference in this trend compared to the short-term forecast is due to the different definitions of the persistence model. At the initial forecasting time steps, deep learning models incorporate additional information from recent hours than persistence models. After 12 hours of forecasting, the FS tends to stabilize.

Table 2: Performance metrics for point-based short-term forecasting under various conditions. rMBD, rRMSE, and FS are in percent (%).

| Models | All | | | | Hard | | | | Easy | | | |
|---|---|---|---|---|---|---|---|---|---|---|---|---|
| | $R^2$ | rMBD | rRMSE | FS | $R^2$ | rMBD | rRMSE | FS | $R^2$ | rMBD | rRMSE | FS |
| Persistence | 0.662 | 2.52 | 38.20 | - | 0.570 | 4.43 | 39.08 | - | 0.695 | 1.76 | 37.52 | - |
| LSTM | 0.709 | 0.78 | 34.29 | 10.23 | 0.627 | **-1.30** | 35.03 | 10.36 | 0.738 | 1.51 | 33.78 | 9.99 |
| LSTM-a | 0.708 | -1.63 | 34.35 | 10.09 | 0.623 | -3.44 | 35.11 | 10.14 | 0.736 | -1.04 | 33.89 | 9.70 |
| Informer | 0.717 | -0.36 | 33.93 | 11.19 | 0.639 | -2.61 | 34.65 | 11.34 | 0.745 | **0.43** | 33.39 | 11.02 |
| Transformer | **0.719** | **0.20** | **33.74** | **11.68** | **0.642** | -1.84 | **34.33** | **12.16** | **0.75** | 0.92 | **33.26** | 11.36 |

Table 3: Performance metrics for point-based long-term forecasting under various conditions. rMBD, rRMSE, and FS are in percent (%).

| Models | All | | | | Hard | | | | Easy | | | | Daily | | | |
|---|---|---|---|---|---|---|---|---|---|---|---|---|---|---|---|---|
| | $R^2$ | rMBD | rRMSE | FS | $R^2$ | rMBD | rRMSE | FS | $R^2$ | rMBD | rRMSE | FS | $R^2$ | rMBD | rRMSE | FS |
| Persistence | 0.455 | -6.42 | 73.08 | - | 0.192 | -47.25 | 100.68 | - | 0.504 | 1.49 | 67.56 | - | 0.445 | 6.98 | 41.68 | - |
| LSTM | 0.608 | -6.54 | 55.83 | 23.61 | 0.445 | -14.28 | 74.31 | 26.20 | 0.650 | -4.28 | 50.32 | 25.52 | 0.534 | 7.32 | 36.78 | 11.75 |
| LSTM-a | **0.672** | **-1.22** | 50.56 | 30.81 | **0.540** | 2.40 | 66.65 | 33.80 | **0.693** | -1.93 | **47.39** | **29.86** | 0.596 | 2.13 | 33.05 | 20.71 |
| Informer | 0.621 | -1.89 | 54.66 | 25.20 | 0.532 | -4.08 | **65.31** | **35.14** | 0.632 | -1.47 | 52.59 | 22.16 | 0.608 | **0.02** | 32.47 | 22.09 |
| Transformer | 0.624 | -1.32 | 54.77 | 25.05 | 0.533 | -4.09 | 65.65 | 34.79 | 0.635 | **-0.78** | 52.66 | 22.06 | **0.617** | 0.60 | **32.18** | **22.79** |

## 5   Remaining Challenges and Other Applications

**Long term discontinuous area-based forecasting.**   Unlike traditional video forecasting or earth system forecasting tasks, solar radiation lacks contiguity during the nighttime. Additionally, the visible bands, which directly correspond to solar radiation, are also unavailable at night. This dataset includes infrared bands to provide continuous cloud information overnight. Investigating how to effectively use this complementary infrared data to assist in long-term solar radiation forecasting during discontinuous periods remains an ongoing area of research.

**Area-based forecasting with sparse ground measurements.**   Similar to weather forecasting datasets, SolarCube provides valuable physics-derived solar radiation using satellite images to enable area-based forecasting. However, ground measurements are only available at the pixels that colocate with the ground monitoring stations (included in the SolarCube). It remains a major challenge

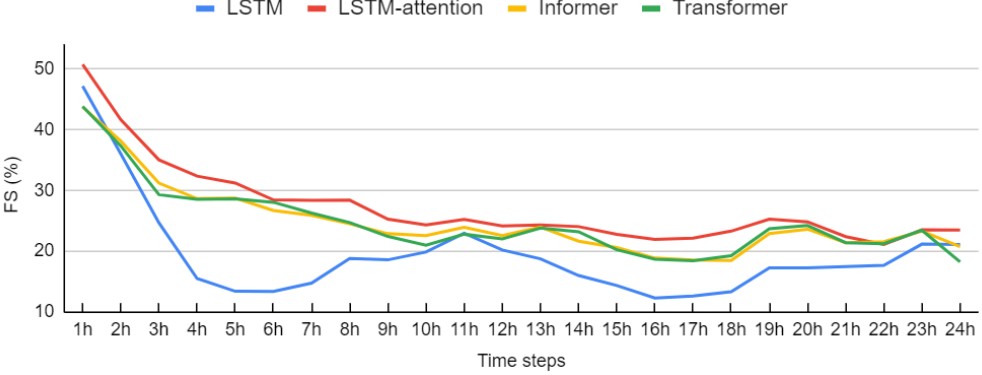

Figure 7: Forecasting skill over time steps for point-based long-term forecasting.

to utilize the very sparse ground measurements over space to enhance area-based forecasting, in particular considering the large degree of variability over space [49; 59]. Physics-guided machine learning may be leveraged to enhance the generalizability beyond the very sparse locations of ground stations [60; 61; 62].

**Fairness considerations in solar radiation forecasting.** The spatial, diurnal, and seasonal variations in solar radiation, along with the fluctuations in energy demand, make it crucial to ensure accurate forecasting of solar radiation for both short and long-term periods. The evaluation metrics and benchmarks developed mainly focus on the prediction quality and have not considered the fairness perspective over geographic regions [63] or the generalization of fairness over time during the forecasting process [64]. Future work is needed to further expand the metrics and benchmarks used for SolarCube on fairness to understand and prevent potential bias incurred by the use of machine learning in solar radiation forecasting.

## 6   Conclusion and Future Work

We introduce a new benchmark dataset SolarCube to meet the growing need for improved solar energy forecasting, promoting more sustainable energy use. The dataset includes satellite images and ground measurements at a 15-minute resolution, along with physics-derived solar radiation using satellite inputs. These datasets enable both point-based and area-based forecasting for lead times ranging from 15 minutes to several days ahead. Auxiliary data are also included to support evaluations under different scenarios (e.g., consistent vs. dynamic cloud coverage, land cover types). Finally, we provide a Python package to allow users to customize the input and output time length, data format, and aggregation level to easily generate data for other variations of the solar radiation forecasting tasks.

**Limitations and future work.** The current version of data collection is uneven globally, with limited ground measurements in less developed and remote regions, as well as a lack of coverage in Europe, where renewable energy demand is high. Furthermore, the dataset is temporally restricted, covering only the year 2018, which may pose challenges for testing the generalizability of the developed models across time. In future work, SolarCube will be expanded to multiple years and include areas covered by the Meteosat Third Generation in Europe. We will also broaden the study area and extend the data range as more minute-level ground measurement sites become available. Additionally, meteorological variables from reanalysis data will be incorporated to enhance long-term forecasting. For model development, we plan to integrate domain-specific fairness metrics into the evaluation framework.

## Acknowledgments and Disclosure of Funding

The work is supported in part by the National Science Foundation under Grant No. 2126474, 2147195, 2425844, 2425845, 2430978, and 2239175, NASA Grant 80NSSC24K1061, Google's AI for Social Good Impact Scholars program, and the Zaratan supercomputing cluster at the University of Maryland.

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
