# SolarCube: Supplementary Information

**Ruohan Li**[1], **Yiqun Xie**[1], **Xiaowei Jia**[2], **Dongdong Wang**[1], **Yanhua Li**[3],
**Yingxue Zhang**[4], **Zhihao Wang**[1], **Zhili Li**[1]
[1]University of Maryland, [2]University of Pittsburgh, [3]Worcester Polytechnic Institute
[4]SUNY - Binghamton
{r526li, xie, ddwang, zhwang1, lizhili}@umd.edu,
xiaowei@pitt.edu, yli15@wpi.edu, yzhang42@binghamton.edu

## A  Data: Access and Datasheet

### A.1  Dataset Access

The SolarCube datasets are available for downloading from Zenodo (https://doi.org/10.5281/zenodo.11498739). The Croissant metadata can be downloaded from Google Drive (https://drive.google.com/drive/folders/1ltOJlLxJk8BxD51LpKD4aSM-TMlAWcza?usp=drive_link).

### A.2  Datasheet

#### A.2.1  Motivations

Solar power has been the fastest-growing power globally, with solar PV installed capacity increasing from 304.3 GW in 2016 to 760.4 GW in 2020 [1]. Studies also suggest that PV will become the dominant electricity supply technology in the cost-optimal climate mitigation scenarios by 2050 [2]. While many studies have explored data-driven methods for solar forecasting, there is a lack of ML-ready datasets for model validation, benchmarking new capabilities, and facilitating the development of better models for various solar forecasting tasks across diverse regions. SolarCube is designed as a multi-purpose dataset to address this gap in the renewable energy sector. With the provided Python package, the datasets can be used for different tasks in solar forecasting, considering forecasting horizons and spatial scale.

A team of researchers with expertise in remote sensing, surface energy budget, and machine learning developed the ML-ready SolarCube. This material is based upon work supported by the National Science Foundation under Grant No. 2105133, 2126474 and 2147195; USGS under Grant No. G21AC10207; Google's AI for Social Good Impact Scholars program; the DRI award and the Zaratan supercomputing cluster at the University of Maryland; and Pitt Momentum Funds award and CRC at the University of Pittsburgh.

#### A.2.2  Composition

SolarCube encompasses 19 study areas across multiple continents including North America, South America, Asia, and Oceania with 9 variables. These include 6 image-based variables and 3 point-based variables. Table 1 lists the variables, their sources, and the spatial-temporal resolution in SolarCube. The image-based variables are organized in a spatio-temporal grid format with three dimensions: latitude, longitude, and time. For each study area, the image-based variables are structured into 600 km by 600 km image patches at a 5 km spatial resolution and feature a year-long image sequence in 2018 with a temporal resolution of 15 minutes, with 35040 time steps in total. A total data points of around 239,673,600,000 are covered for each image-based variable. A visualization of the

image-based variables is shown in Figure 1. The point-based variables are located over 19 ground measurement sites within the 19 study areas. The measured solar radiation is provided with 1 min temporal resolution in 2018, which contributes to around 9986400 data instances. The rest of the point-based variables are static for model evaluation scenarios.

The three satellite bands variables (vis047, vis086, and ir133), solar zenith angle (SZA), satellite-derived solar radiation (SSR), and ground-measured solar radiation are used for model training. The rest of the variables are mainly used for model evaluation. Users can define their features and target variables based on their specific tasks. In this study, for area-based forecasting tasks, the label is SSR. For point-based forecasting tasks, the label is ground-measured solar radiation. For short-term forecasting, each sample contains continuous data within 24 time frames, sampled with a stride of 8 time steps. For long-term forecasting, each sample contains continuous data within 192 time frames, sampled with a stride of 72 time steps. All these sampling choices can be customized using the provided Python package available at https://github.com/Ruohan-Li/SolarCube.

The 19 study areas are a subset of the total areas covered by the GOES-16 and Himawari-8. The selection of the study areas is mostly restricted by the availability of ground monitoring sites that provide minute-wise solar radiation measurement. We include all available sites that fulfill the requirements. We provide information about the sites and their corresponding study areas in Table 2. The location of the study area and the diversity of landscapes are summarized in the main text. To make the data size of SolarCube manageable while still meeting the needs of various forecasting horizon tasks, we have selected study areas with a spatial scale of 600 km by 600 km. Except for ir133, the other dynamic variables are only available during the daytime due to the nature of the sun. This study only focuses on solar radiation forecasting over terrestrial, hence the ocean part of the imaged-based variables are excluded. The ground measurement instrument may occasionally malfunction or produce suspicious measurements that do not pass the quality check due to various reasons (e.g., dust cover). Such data are excluded. The availability of all variables is summarized in separate files provided along with the data.

We use the data of 14 study areas as training and test the rest of 5 study areas. The choice of the testing study areas is also listed in Table 2. We split the dataset in this way to test ML performance in large-scale applications. Testing on independent study areas can better indicate the models' generalizability over other regions considering spatial variability [3].

Table 1: Table of all variables in SolarCube

| Variable | Source | S. Res. | T. Res. | Ref. |
|---|---|---|---|---|
| **Area-based variables** | | | | |
| 0.47μm visible channel of GOES-16 and Himawari-8 (vis047) | GeoNEX | 5km | 15min | [4] |
| 0.86μm visible channel of GOES-16 and Himawari-8 (vis086) | GeoNEX | 5km | 15min | [4] |
| 13.3μm infrared channel of GOES-16 and Himawari-8 (ir133) | GeoNEX | 5km | 15min | [4] |
| Solar Zenith Angle (sza) | GeoNEX | 5km | 15min | [4] |
| Satellited derived Solar Radiation (ssr) | - | 5km | 15min | |
| Cloud Mask (cm) | NOAA & EUMETSAT NWC SAF | 5km | 15min | [5] [6] |
| **Point-based variables** | | | | |
| Ground-measured solar radiation | SURFRAD, BSRN | point | 1min | [7] [8] |
| Land surface types | MODIS | point | static | [9] |
| Elevation | GTOPO30 | point | static | [10] |

### A.2.3 Collection Process

Except for the satellite-derived solar radiation variables, all other variables are obtained from public datasets available on their respective official websites. These official datasets come with user guides that document the data quality and validation results, as listed in Table 1. These variables were collected by the authors through direct downloads from the websites. No crowdworkers were involved in the data collection process, and no ethical review was conducted.

**Satellite Derived Solar Radiation Data Validation** SSR is the only variable not directly obtained from existing datasets. However, the methodology for deriving this data is well-established and has been successfully used to generate numerous public official datasets, including NASA

Table 2: Table of the study area and ground measurement site. $lat\_ulcnr$, $lon\_ulcnr$, $lat\_lrcnr$, and $lon\_lrcnr$ represent the upper left corner latitude, upper left corner longitude, lower right corner latitude, and lower right corner longitude, respectively.

| Id | Name | Latitude | Longitude | Network | Elevation | Timezone | lat_ulcnr | lon_ulcnr | lat_lrcnr | lon_lrcnr | Test |
|---|---|---|---|---|---|---|---|---|---|---|---|
| 1 | bon | 40.050 | -88.370 | SURFRAD | 213.0000 | America/Chicago | 43.05 | -91.37 | 37.05 | -85.37 | no |
| 2 | fpk | 48.310 | -105.100 | SURFRAD | 623.3125 | America/Denver | 51.31 | -108.10 | 45.31 | -102.10 | no |
| 3 | gwn | 34.250 | -89.870 | SURFRAD | 101.0625 | America/Chicago | 37.25 | -92.87 | 31.25 | -86.87 | no |
| 4 | dra | 36.620 | -116.020 | SURFRAD | 998.0625 | America/Los_Angeles | 39.62 | -119.02 | 33.62 | -113.02 | no |
| 5 | psu | 40.720 | -77.930 | SURFRAD | 375.5625 | America/New_York | 43.72 | -80.93 | 37.72 | -74.93 | yes |
| 6 | sxf | 43.730 | -96.620 | SURFRAD | 476.3125 | America/Chicago | 46.73 | -99.62 | 40.73 | -93.62 | no |
| 7 | tbl | 40.120 | -105.240 | SURFRAD | 1651.5625 | America/Denver | 43.12 | -108.24 | 37.12 | -102.24 | yes |
| 8 | FLO | -27.533 | -48.517 | BSRN | 55.0000 | America/Sao_Paulo | -24.00 | -54.00 | -30.00 | -48.00 | no |
| 9 | LRC | 37.104 | -76.387 | BSRN | 4.2500 | America/New_York | 42.00 | -78.00 | 36.00 | -72.00 | yes |
| 10 | ASP | -23.798 | 133.888 | BSRN | 548.0625 | Australia/Darwin | -18.00 | 132.00 | -24.00 | 138.00 | no |
| 11 | COC | -12.193 | 96.835 | BSRN | 3.0625 | Indian/Cocos | -12.00 | 96.00 | -18.00 | 102.00 | yes |
| 12 | DWN | -12.424 | 130.893 | BSRN | 25.9375 | Australia/Darwin | -12.00 | 126.00 | -18.00 | 132.00 | no |
| 13 | FUA | 33.582 | 130.375 | BSRN | 8.0625 | Asia/Tokyo | 36.00 | 126.00 | 30.00 | 132.00 | no |
| 14 | HOW | 22.554 | 88.306 | BSRN | 6.5000 | Asia/Kolkata | 24.00 | 84.00 | 18.00 | 90.00 | no |
| 15 | ISH | 24.337 | 124.163 | BSRN | 11.5000 | Asia/Tokyo | 30.00 | 120.00 | 24.00 | 126.00 | no |
| 16 | LAU | -45.045 | 169.689 | BSRN | 352.5000 | Pacific/Auckland | -42.00 | 168.00 | -48.00 | 174.00 | no |
| 17 | NEW | -32.884 | 151.729 | BSRN | 19.6875 | Australia/Sydney | -30.00 | 150.00 | -36.00 | 156.00 | no |
| 18 | SAP | 43.060 | 141.328 | BSRN | 20.5625 | Asia/Tokyo | 48.00 | 138.00 | 42.00 | 144.00 | yes |
| 19 | TAT | 36.058 | 140.126 | BSRN | 28.1250 | Asia/Tokyo | 42.00 | 138.00 | 36.00 | 144.00 | no |

MODIS/Terra+Aqua Surface Radiation product (MCD18) [11] and the GeoNEX DSR/PAR Surface Radiation product [12]. We used the same data sources and methodologies as the GeoNEX DSR/PAR dataset [12]. The input variables are listed in Table 3. The method is elaborated in Section 3 of the main manuscript.

To confirm the quality of the SSR in SolarCube, we conduct a comprehensive comparison with existing datasets. We validate the SSR by comparing them with the ground measurements over the 19 sites. The evaluation metrics are similar to the forecasting tasks, which are $R^2$, RMSE, MBD, and their relative values for fair comparison. The validation results of SolarCube raw temporal resolution (15min) are listed in Table 4. There are no other image-scale solar radiation products with the same temporal resolution covering the study areas for comparison. The highest temporal resolution of the current image-scale solar radiation datasets with the same coverage as the proposed studies is 1 hour. Therefore, we aggregate the SSR of SolarCube to 1 hour to facilitate better comparison with these datasets. The hourly SSR of SolarCube is obtained by calculating the mean of the available 15-minute resolution data for that hour. The hourly ground-measured solar radiation is averaged from all available 1-minute resolution data within that hour. The datasets compared in this study include both satellite-derived and reanalysis data. Their corresponding spatial and temporal resolutions and the comparison of the dataset validation results are shown in Table 4. SolarCube presents a similar accuracy to the GeoNEX dataset at an hourly scale [12], further demonstrating the robustness of the methods. SolarCube demonstrates significantly better performance compared to other datasets, including the benchmark satellite-derived datasets CERES, the newly developed satellite-derived datasets EPIC, and reanalysis data ERA5. Additionally, we plot a scatter plot to compare SolarCube with ERA5, which is widely used in earth system forecasting, as elaborated in Section 2 of the main text. The results are shown in Figure 2, where the color of the scatterplot represents the sample density.

Table 3: Data sources and their spatial and temporal resolution in generating satellite-derived solar radiation data

| Input Variable | Source | S. Res. | T. Res. |
|---|---|---|---|
| 0.47μm visible channel of GOES-16 and Himawari-8 | GeoNEX | 1km | 10/15min |
| solar zenith angle, sensor zenith angle, relative azimuth angle | GeoNEX | 1km | 10/15min |
| Surface albedo | MODIS, Climatology | 1km | Daily |
| Total precipitable water vapor | MERRA2 | $0.5 \times 0.625°$ | Hourly |
| Surface elevation | GTOPO30 | 30 arcsec | Static |

## A.2.4 Preprocessing, cleaning, labeling

The satellite data (including three-band and SZA variables) from GOES-16 are downloaded at 15-minute intervals, while data from Himawari-8 are downloaded at 10-minute intervals. Both datasets

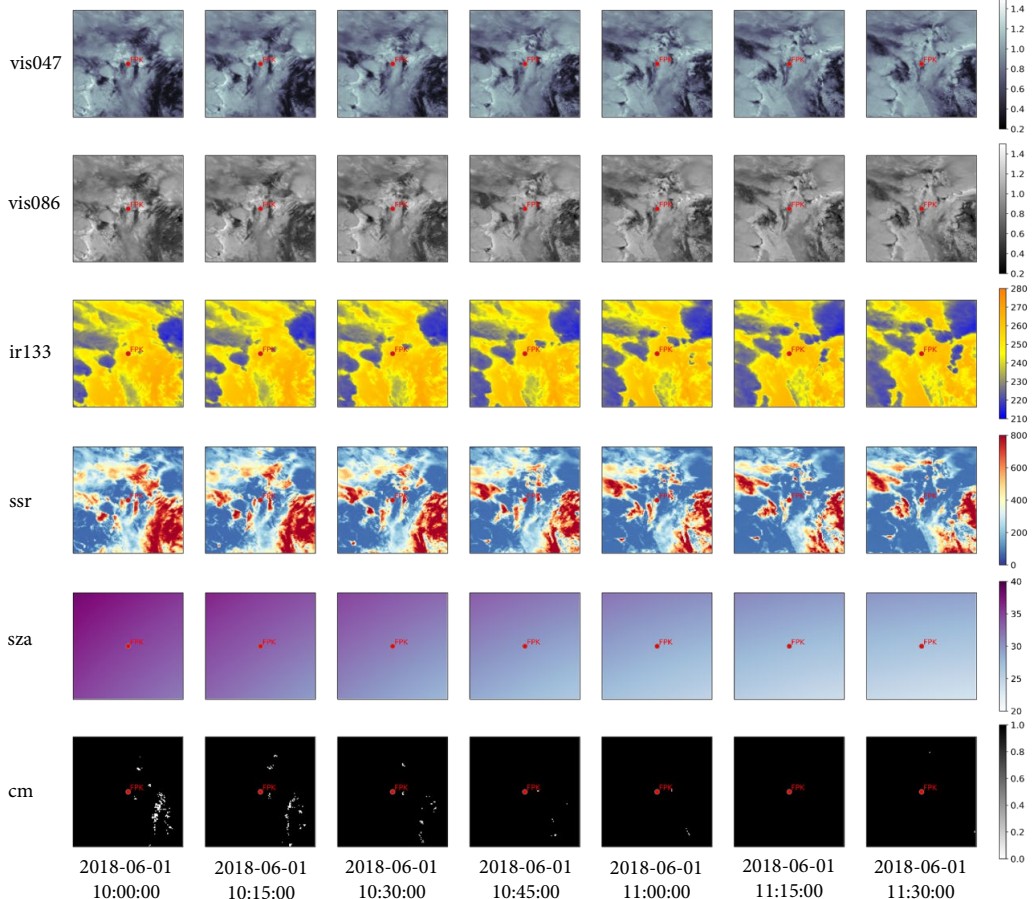

Figure 1: Visualization of all image-based variables for a time sequence

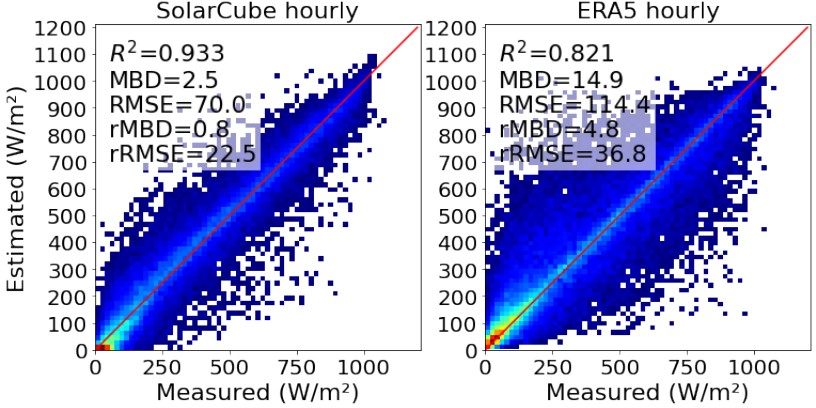

Figure 2: Comparasion of SolarCube SSR and ERA5 at hourly scale

Table 4: Validation results of satellite-derived solar radiation variables in SolarCube and other image-scale solar radiation datasets

| Datasets | Type | Temp. Res. | Sp. Res. | Metrics | | | | |
|----------|------|-----------|----------|---------|-----|------|------|-------|
| | | | | $R^2$ | MBD | RMSE | rMBD | rRMSE |
| SolarCube | Satellite-derived | 15min | 5km | 0.904 | 3.4 | 94.3 | 1.0 | 26.7 |
| SolarCube | Satellite-derived | hourly | 5km | 0.933 | 2.5 | 70.0 | 0.8 | 22.5 |
| ERA5 | Reanalysis | hourly | 0.5° ( 50km) | 0.821 | 14.9 | 114.4 | 4.8 | 36.8 |
| CERES | Satellite-derived | hourly | 1° ( 100km) | 0.903 | 3.2 | 82.6 | 1.0 | 26.6 |
| EPIC | Satellite-derived | hourly | 0.1° ( 10km) | 0.804 | 13.4 | 119.2 | 4.3 | 38.3 |

have a raw spatial resolution of 1 km. To ensure temporal consistency, we averaged two observations from Himawari-8 around each 15-minute mark (e.g., using 00:10 and 00:20 to calculate 00:15). We used the time-aligned satellite data to produce the SSR variables, ensuring the same temporal and spatial resolution, and aligned them well in a datacube. The cloud mask datasets for GOES-16 and Himawari-8 are structured in a Cartesian coordinate system. We extracted the center latitude and longitude of each pixel in the datacube and applied a projection transformation to find the corresponding pixel in the cloud mask products. The values of the cloud mask were then assigned to the datacube as well. We then aggregated the datacube from a 1 km resolution to a 5 km resolution to create a more portable dataset. The rationale for choosing a 5 km resolution is elaborated in the main text. All the image-based variables are structured in the datacube with dimensions of 35040x600x600, representing time, height, and width. For point-based variables, the preprocessing and cleaning of the ground measurements are described in the main text. The land cover types and elevations are extracted from their raw dataset based on the latitude and longitude of the sites. All variables are preprocessed using Python packages such as netCDF4 and h5py.

### A.2.5 Uses

The SolarCube has been used for three subtasks presented in the main text, including area-based short-term forecasting, point-based short-term forecasting, point-based long-term forecasting. Additional tasks that can be used are listed in Section 5 in the main text. The current dataset composition sufficiently supports a wide range of solar radiation forecasting tasks. However, for ultra-long term forecasting (longer than 24h), additional weather data and a larger image spatial scale would be beneficial. In the next release, we plan to include these extra variables and provide a low-resolution version of image-based variables with a larger spatial scale for ultra-long-term forecasting. All updates to the next release will be documented and made available on the dataset and code webpage.

### A.2.6 Distribution

The SolarCube is an open dataset and will distributed through Zenodo https://doi.org/10.5281/zenodo.11498739 with a Creative Commons Attribution 4.0 International License. Users can freely download it without restrictions. The Python package which allows users to customize the input and output time length, data format, and aggregation level to generate data for other variations of the tasks is also shared in https://github.com/Ruohan-Li/SolarCube.

### A.2.7 Maintenance

The University of Maryland will support, host, and maintain the dataset. The managers of the dataset can be contacted through the following emails: Ruohan Li (r526li@umd.edu) and Yiqun Xie (xie@umd.edu). There is no erratum. If errors are found in the future or additional features are added, the dataset will be updated and released as a new version on Zenodo. Corresponding announcements will be posted on the project's GitHub page. In the meantime, older versions of the dataset will continue to be maintained and hosted. Currently, there is no mechanism for others to extend, augment, build on, or contribute to the dataset.

### A.2.8 Author Statement

The authors of this paper bear all responsibility for any violation of rights and confirm that the data is properly licensed.

## B  Models: Access and Additional Details

### B.1  Code Access

The SolarCube Python package for sampling and visualizing data, along with the benchmark model, can be accessed at https://github.com/Ruohan-Li/SolarCube

### B.2  Additional Details

The configurations of the benchmark models used for different tasks are summarized in this section. For all tasks, we generate the predictions in a non-auto-regressive way.

### B.2.1  Track 1: Area-based forecasting.

**ConvLSTM**   We adopted the default configuration of the ConvLSTM model in https://github.com/jhhuang96/ConvLSTM-PyTorch.git. The model is structured in a encoder-decoder architecture with leaky ReLU activations. The encoder starts with three convolutional layers, each with a kernel size of 3×3. Following these, three ConvLSTM cells are employed to handle the temporal dimension, each with a filter size of 5. The decoder mirrors this structure in reverse, beginning with ConvLSTM cells to manage the temporal data and then utilizing deconvolutional layers to upsample the spatial dimensions. The kernel size for the first two deconvolutional layers is 4×4, while the last layers use a combination of 3×3 and 1×1 kernels. The initial learning rate is set as 0.0001 and is dynamically reduced by half if the validation performance does not improve for 4 consecutive epochs.

**Space-time Transformer Models**   We consider three variants of space-time transformers: (1) the axial attention model (Axial), (2) the video swin-transformer (Video-swin), and (3) the divided space-time attention model (Divided-st), which are based on EarthFormer (https://github.com/amazon-science/earth-forecasting-transformer.git). We use the default configuration in the EarthFormer. Earthformer employs a hierarchical encoder-decoder architecture. Each hierarchy stacks four cuboid attention blocks. Several cuboid attention layers for different cuboid attention patterns are enclosed in each block. The three variants differ from each other by the choice of the cuboid attention patterns in the encoder, which are summarized in Table 5.

(1) For Axial, there are three cuboid attention layers, each separated along a different dimension: temporal, width, and height. There is no window shift offset when separating the cuboids.

(2) For Video-Swin, two cuboid attention layers are included. Both layers have a cuboid pattern with dimensions (2, 4, 4). One layer is separated without a window shift, while the other is separated with a window shift of half the cuboid size along each dimension.

(3) Divided-ST also has two cuboid attention layers. One layer is separated along the temporal dimension, and the other along the spatial dimensions. To save memory, we use half the size of the spatial dimensions as one cuboid size.

When multiple cuboid attention layers are stacked, each one is paired with layer normalization and a feed-forward network. The decoder uses the "Axial" pattern for its cuboid blocks. To adjust the spatial resolution before applying the cuboid attention layers, the model integrates initial downsampling and upsampling modules. The downsampling layer consists of one 2D convolutional layer and one patch-merge layer, which halves the spatial scale and merges the spatial dimensions into channels. The upsampling modules are composed of one nearest neighbor interpolation layer. The final model is trained with an initial learning rate of 0.001, using a cosine annealing schedule, gradient clipping at 1.0, and a warmup phase over the first 20% of the 100 epochs.

Table 5: Configurations of the cuboid attention patterns of Axial, Video-swin, and Divided-st. $T$, $H$, and $W$ represent the time sequence length, height, and width of the input tensor. Shift represents the window shift offset when separating the cuboids [13].

| Model Name | Configurations | Values |
|---|---|---|
| Axial | cuboid_size | $(T,1,1) \rightarrow (1,H,1) \rightarrow (1,1,W)$ |
| | shift | $(0,0,0) \rightarrow (0,0,0) \rightarrow (0,0,0)$ |
| Video-swin | cuboid_size | $(2,4,4) \rightarrow (2,4,4)$ |
| | shift | $(0,0,0) \rightarrow (1,2,2)$ |
| Divided-st | cuboid_size | $(T,1,1) \rightarrow (1,H/2,W/2)$ |
| | shift | $(0,0,0) \rightarrow (0,0,0)$ |

#### B.2.2 Track 2: Point-based forecasting

**LSTM**    The LSTM model is composed of two LSTM layers with 128 neurons each. These layers are followed by a linear layer with 64 neurons and a final output layer. The model is trained with a learning rate of 0.001.

**LSTM-attention**    The LSTM-attention model follows an encoder-decoder architecture, where an LSTM encoder processes the input sequence and generates a sequence of hidden states, and an LSTM decoder takes the hidden states as the initial states and generates the output sequence. The attention scores are computed using the Bahdanau-style (additive) attention from the encoder outputs and then applied to the decoder outputs, followed by a linear output layer. The inputs of the encoder and decoder are embedded with 256 neurons respectively. Both LSTM layers have 256 neurons. The model is trained with a learning rate of 0.001.

**Informer**    We implemented the Informer model following https://github.com/zhouhaoyi/Informer2020.git with the default configuration. Informer has a ProbSparse self-attention mechanism to enhance efficiency. The input data is embedded with an output dimension of 512. The model includes 2 encoder blocks and 1 decoder block, composed of self-attention and feed-forward layers. The feed-forward layer dimension is set to 2048. The multi-head attention mechanism is configured with 8 heads. GELU is used as the activation function. The label length is set to match the length of the input sequence. The learning rate follows a one-cycle schedule, increasing to a maximum of 0.0001.

**Transformer**    The model shares the same settings as the Informer for the number of encoder layers, decoder layers, embedding layer dimension, feed-forward layer dimension, the number of heads, the activation function, and the learning rate. The only difference is that it uses the regular self-attention instead of the ProbSparse self-attention in Informer.