# OpenReview forum: "SolarCube: An Integrative Benchmark Dataset Harnessing Satellite and In-situ Observations for Large-scale Solar Energy Forecasting"
_NeurIPS.cc/2024/Datasets_and_Benchmarks_Track — NeurIPS 2024 Track Datasets and Benchmarks Poster_

### Official Review · Reviewer_qt78 · 2024-07-22
**Review of 1331**

**Rating:** 7
**Confidence:** 4
**Correctness:** Yes, I think the dataset and benchmar…
**Clarity:** Largely yes, but please see comments …

**Review:**

Overall this is a clear paper that makes a good contribution to the literature.

On the positives, the paper and dataset are clearly motivated and well-explained. There are lots of things that have been done right.

I have some smaller concerns, which I've listed in the "Opportunities for Improvements". These are somewhat small and would improve the paper. However, they're not show-stoppers.

**Strengths:**

+ The task is clearly motivated and the paper makes a good connection from the dataset to the task.

+ The methods and evaluation are comprehensive and the authors have been thoughtful in their setup. For example, the persistence model that is being compared with is well-done and the authors worked to make it work better than naive persistence. Similarly, the authors split by location and explain why splitting by time is a poor idea. These sorts of details matter in dataset papers and the authors have clearly thought through the dataset .

+ The challenge and data are well-explained.

**Additional Feedback:**

None

**Documentation:**

The documentation is sufficiently clear.

**Ethics:**

I have no ethical concerns.

**Limitations:**

Yes, I think the limitations are adequate.

**Opportunities For Improvement:**

These are primarily nitpicks aimed at making the paper better.

- Some of the data prep could be explained better. If I understand correctly, the ground-truth for the method comes from another method. This is entirely ok -- "ground-truth" 3D is usually the result of a stereo or laser system. However, since this method has known errors, it would be great to integrate this into the evaluation somehow. What's the relative size of the the mistakes being made in forecasting?

- It's not clear to me that presenting only the rRMSE / rMBD makes sense. I understand the goal to normalize across sites and perhaps there's a way to do it that way? However, if it's point-wise normalized, this seems to considerably upweight performance close to dusk/dawn when these times presumably have considerably less impact practically than performance near noon.

- There are a few surprising examples that might benefit from further looking at. Perhaps there are interesting insights to be gleaned. (a) In Table 1  It's surprising that the hard cases appear to be "easier" for area forecasting in a few of the metrics. For the skill score, this might be that the persistence model is worse, and so the skill enabled from learning goes up. But for rRMSE this is a bit surprising. (b) In Table 3, LSTM-a's performance mainly comes from improvements on the easy cases. Is there something interesting going on here? I'd have assumed that the transformer would have similar performance.

- While there are differences between the methods, many of the methods are quite similar. I'd be hard pressed to come up with a "clear winning" model in many cases. The paper may want to report the measurement error of the data or the error with respect to these metrics. This'll help identify between significant differences.

Small stuff:
- Please expand out "MBD" as "Mean Bias Deviation", if this is what it is. The paper's pretty compressed already, but it might be good to have definitions in the supplement since MBD won't be familiar to most NeurIPS readers.
- I haven't seen skill scores reported as percentages. Since skill scores can be negative (worse performance than the baseline), it's a bit odd to interpret skill scores as a percentage. It may be good to report the FS for persistence as 0 rather than -.
- L240: "cleaners index" -> "clearness index"
- Figures 2, 3, and 4 would look much better as pdfs. Figure 1 has had the images converted to jpgs and would look better if the underlying data were png.

**Relation To Prior Work:**

To the best of my understanding, the paper describes the related work well and draws clear distinctions.

**Summary And Contributions:**

The paper proposes a dataset for solar energy forecasting. The underlying data covers 19 study areas and presents three benchmarks: (1) image-based (i.e., with spatial data) short-term; (2) point-based (i.e., just a temporal signal at a location) short-term; and (3) point-based long-term. The paper benchmarks multiple different methods and presents metrics for evaluation.

---

> ### Author Rebuttal · Authors · 2024-08-17
>
> We appreciate your constructive comments for improving the paper, and please find our responses below.
>
> __1.Some of the data prep could be explained better. If I understand correctly, the ground-truth for the method comes from another method. This is entirely ok -- "ground-truth" 3D is usually the result of a stereo or laser system. However, since this method has known errors, it would be great to integrate this into the evaluation somehow. What's the relative size of the the mistakes being made in forecasting?__
>
> Thanks for the question and suggestion. For area-based forecasting, we use physics-derived solar radiation from satellite images as the ground truth reference. Based on the retrieval error in Fig. 2 of the supplementary file, the relative size of the errors (absolute value of ML prediction error / Data error) are about 6.4, 4.1, 5.4, and 3.5 for ConvLSTM, Axial, Video-swin and Divided-st, respectively, using rMBD as an example. We will incorporate these results into the supplementary document and reference them in the main paper.
>
> __2.It's not clear to me that presenting only the rRMSE / rMBD makes sense. I understand the goal to normalize across sites and perhaps there's a way to do it that way? However, if it's point-wise normalized, this seems to considerably upweight performance close to dusk/dawn when these times presumably have considerably less impact practically than performance near noon.__
>
> Thank you for your feedback. We would like to clarify that the rRMSE and rMBD metrics are calculated as RMSE/mean(true), so they do not involve point-wise normalization. As a result, the performance near dusk/dawn is not specially weighted. We have now added this clarification to avoid confusion caused by the ambiguity.
> These metrics are used to facilitate better comparisons across different studies. Based on the suggestion, we will also add the original RMSE and MBD values in the supplementary materials to provide more comprehensive assessment results.
>
> __3.There are a few surprising examples that might benefit from further looking at. Perhaps there are interesting insights to be gleaned. (a) In Table 1 It's surprising that the hard cases appear to be "easier" for area forecasting in a few of the metrics. (b) In Table 3, LSTM-a's performance mainly comes from improvements on the easy cases. Is there something interesting going on here?__
>
> We appreciate the reviewer's insightful comments. Regarding (a), for rRMSE and rMBD, their values reported in Table 1 are lower for the hard cases are mainly caused by their higher mean values (because of typically clearer-sky conditions) when calculating the relative scores. Given this, we recommend placing more emphasis on R² when comparing hard and easy scenarios, and we will highlight this in the paper.
>
> For (b), this is indeed an interesting observation. We found that both LSTM and LSTM-a exhibit higher FS for the easy cases, whereas Transformer and Informer show better FS for the hard cases in long-term forecasting. For the easy cases, our interpretation is that this might be caused by the trade-off between model size and generalizability. The smaller size LSTM-a might be sufficient for the easy cases but limited for the hard cases, and its performance may generalize better on test data. On the other hand, the more complex transformer-based models may have better ability for hard cases but do not generalize as well to test data. As a result, for easy cases, LSTM-a may have an advantage due to its sufficient ability and better generalizability on test data. We feel this could be an interesting topic for future research, and will add a discussion in Section 5 to recommend further studies to have more detailed analysis on the suitability of different models under different scenarios.
>
> __4.While there are differences between the methods, many of the methods are quite similar. I'd be hard pressed to come up with a "clear winning" model in many cases. The paper may want to report the measurement error of the data or the error with respect to these metrics. This'll help identify between significant differences.__
>
> Thanks for the suggestion. We notice that the differences between models are relatively smaller in particular for short-term point-based forecasting. For this task, the measurement error of the observations is around 2%~5%. Our interpretation is that this short-term task is in general easier due to temporal autocorrelation, making the differences smaller across methods. We will report these context information to the model comparisons. The differences are relatively larger between different categories of models in area-based forecasting and long-term point-based forecasting. For area-based forecasting, the major differences are in the hard cases, and the gaps are reduced for easy cases. We also observe that the differences between different categories of models (e.g., ConvLSTM and transformer-based models) are larger than the differences between different transformer-based models. We will also add this to the result discussion.
>
> __5. Please expand out "MBD" as "Mean Bias Deviation", if this is what it is.__
>
> We will define Mean Bias Deviation (MBD) and Root Mean Square Error (RMSE) in the supplement.
>
> __6.I haven't seen skill scores reported as percentages. Since skill scores can be negative (worse performance than the baseline), it's a bit odd to interpret skill scores as a percentage. It may be good to report the FS for persistence as 0 rather than -.__
>
> We will change all FS to decimal and change persistence to 0.
>
> __7.L240: "cleaners index" -> "clearness index"__
>
> We will revise "cleaners index" to "clearness index".
>
> __8.Figures 2, 3, and 4 would look much better as pdfs. Figure 1 has had the images converted to jpgs and would look better if the underlying data were png.__
>
> We have re-exported Figures 2, 3, and 4 as PDFs and updated the underlying images in Fig. 1 to PNG for better visualization quality.

---

### Official Review · Reviewer_AUxG · 2024-07-24
**A great new dataset for an important problem**

**Rating:** 7
**Confidence:** 4
**Correctness:** yes
**Clarity:** the paper is very well constructed.

**Review:**

There are very few high-quality datasets available for solar irradiance. This work alleviates that problem to a great degree. The authors have done a fantastic job of using publicly-available satellite imagery data, and on ground in-situ observational network to build this comprehensive dataset. The temporal and spatial resolutions are very optimal for operationalizing models built using this data. This will provide an important tool to advance solar forecasting using ML.

**Strengths:**

The key strength of this work is in the novelty of fusing data streams from satellites to ground observations providing an orthogonal dataset for solar irradiance to what is out there currently.

**Additional Feedback:**

n/a

**Documentation:**

yes

**Limitations:**

Since the work heavily relies on satellites and ground sensing network, it maybe limited by the availability of such data in the Global South.

**Opportunities For Improvement:**

Can the authors comment on the

- spatial/geographical inequity in the quality of data. If so, how easily can this be extended to other parts of the world?

- quality checks for ground observational irradiance data.

- any other biases present in the data?

**Relation To Prior Work:**

yes

**Summary And Contributions:**

the authors present a new data source for modeling solar irradiance, a key weather diagnostic variable that enables forecasting solar energy, reducing the associated uncertainties around cloud cover.

---

> ### Author Rebuttal · Authors · 2024-08-17
>
> We appreciate your constructive comments for improving the paper, and please find our responses below.
>
> __1.spatial/geographical inequity in the quality of data. If so, how easily can this be extended to other parts of the world?__
>
> Thanks for the question, which is an important consideration when adapting the model to different geographic regions. We have expanded the discussion in Sec. 5 to include a task that explicitly targets cross-region evaluation, in which the train-test split is based on disjoint geographic regions. This will help better understand a model’s cross-region generalizability under different scenarios (e.g., from warm to cold climates; from plains to mountainous regions). In addition, we added discussions on integrating knowledge-guided learning methods, where physical rules represented by variable relationships could potentially help better regularize the training and enhance the generalizability over space.
>
> __2. quality checks for ground observational irradiance data.__
>
> The ground observational irradiance data is measured by pyrheliometers and pyranometers provided by the Baseline Surface Radiation Network (BSRN) and the Surface Radiation Budget Network (SURFRAD). The instrument operating errors are reported around  2% for pyrheliometers and ~5% for pyranometers – or 15 Watts/m2 (https://gml.noaa.gov/grad/surfrad/overview.html). All data have passed the quality check set by the BSRN standard offered by the BSRN-Toolbox. The flagged data are properly handled following guidelines in [1]. We will expand this information in the in-situ data processing part in Section 3.
>
> [1]https://doi.org/10.5194/amt-4-339-2011
>
> __3.Any other biases present in the data?__
>
> Thanks for the question. One thing we can think about is that similar to many other applications, the availability of the ground measurement data is relatively more limited in less developed regions and places that are hard to reach or build stations. The combination of satellite observations, which has broader coverage, and ground measurements could be one path forward to mitigate this. This is also related to the next comment, where we will provide more details. We will add this information to the discussion in Section 5.
>
> __4.Since the work heavily relies on satellites and ground sensing network, it maybe limited by the availability of such data in the Global South.__
>
> Yes, the global south does have comparatively lower availability. The limitation is mainly on the measurements from ground stations. The availability of satellite observations are more similar as geostationary satellites cover about the same areas for global south and global north. Thanks for pointing this out, and we have added the Global South consideration in the discussion, which should be considered for future network expansion. We will also regularly check new developments about data collection in Global South in future work and integrate the new data as they become available.

---

### Official Review · Reviewer_aSBK · 2024-07-25
**First review of SolarCube**

**Rating:** 8
**Confidence:** 2

**Review:**

The submitted work is highly relevant as it provides a multi-modal benchmark dataset for an important application, namely the prediction of solar radiation, which has a direct impact on the efficient integration of renewable energy sources into energy grids.

The authors motivate their work describing the gap in prior work which either features datasets of ground-based sky images that have a very limited local scope (point-based forecasting), and or datasets covering more geographic extent but not focused on solar radiation application. Hence, the dataset bridges a gap between point-based forecasting and area-based forecasting which is important for large-scale solar power system management. Furthermore, providing year-long time-series with 15-minutes intervals, the dataset serves short-term and long-term applications.

After describing the data curation process, the authors define different forecasting tasks, namely area-based and point-based forecasting with different time horizons.

I appreciate the clarity/honesty of the report, for example it is explicitly stated that area-based long-term forecasting is not tested due to limitations provided in the literature, yet the framework allows researchers to experiment.

Furthermore, different testing scenarios are established including easy scenarios with steady cloud cover and hard scenarios with dynamic cloud cover. This provides a test environment which allows to evaluate the behavior of the models under realistic circumstances.

**Strengths:**

The presented dataset is of high relevance to the research community within the domain of solar energy forecasting as it integrates different data modalities which haven’t been combined in such a systematic and large-scale before.

Furthermore, the definition of different scenarios with regards to cloud covers and the comparison with the baseline persistence model which assumes steady atmospheric conditions, allows for model evaluation under realistic circumstances which is of direct practical implications.

**Additional Feedback:**

Abstract: Detail short- and long-term

Line 199: Typo: insufficient -> sufficient

Supp. Material: Empty C Additional results

**Clarity:**

Overall, the paper follows a clear structure and is written in a detailed and concise way.

Please review the points I mention in the above “Opportunities For Improvement” section to further improve clarity.

**Correctness:**

The presented datasets are curated in a sound way. The collection and preparation process are described in high detail in the paper. Different performance metrics help to evaluate and compare different skills of the models.

**Documentation:**

A datasheet is comprehensively answered in the Supplementary Material.

At the time of submission, the datasets are provided through a google drive link. However, the authors allocated already a Zenodo repo with DOI where they will release the dataset under an open license before publication.

The code is available on a github repo. The license of the codes is missing, both on the github repo and in the Supp. Material.

**Ethics:**

No ethical concerns are observed.

**Limitations:**

Limitations are discussed in detail, in particular with regards to data sparsity (spatial) and discontinuity (temporal) as well as a brief discussion on fairness and geographic bias (though this could be extended as mentioned above).

**Opportunities For Improvement:**

I would appreciated a more detailed discussion (challenges and opportunities) on the distinct behavior of different geographic regions and how insights from one region can be leveraged in other regions (aka transfer learning). The authors touch upon this point just before the final conclusion section, but it would be interesting if they could put some hypotheses.

The problem formulation can be improved and described more clearly. What are the input features and what are considered ground-truth labels in this ML-modeling set-up?

Persistence model appeared ad-hoc, introduce it better, especially for people not familiar with the field

Can you introduce MBD in the text (maybe also RMSE for consistency).
In Figure 2, the title of the middle graph is rBias. Should this be rMBD?

Table 1: Please indicate that values are in percent (%).

One thing I don’t quite understand is why the video-swin shows the lowest bias by far in the experiment under different land cover types (Fig. 2), but doesn’t show generally low bias under various cloud conditions in Table 1. Can you discuss on this point?

In Line 304, you describe “The LSTM-a model shows lower FS for the initial time steps but demonstrates a more pronounced increase in FS as the time step advances.” I don’t conclude the same when looking at the graph, in the very early time steps, the other models have similarly low FS value. I would rather conclude that the FS score is more variable compared to the curves of the other models?!

**Relation To Prior Work:**

Previous work is comprehensively discussed with regards to forecasting approaches and existing datasets. Limitations of the prior work is illustrated (e.g. SEVIR which is focused on precipitation/storm forecasting and doesn’t contain ground observation, hence not suited for solar radiation forecasting), motivating the work of the submission.

**Summary And Contributions:**

The authors present a new benchmark dataset for solar energy forecasting integrating (geostationary) satellite imagery, satellite-derived solar radiation data and ground-based observation as well as auxiliary context data such as land cover and cloud masks.
The authors claim that the created dataset is able to bridge a gap between point-based and area-based as well as short-term and long-term solar radiation forecasting.
The submission features the description of the dataset curation, from collection and pre-processing up to dataset split to provide a ML-ready dataset.
Finally a benchmark task is defined based on which several models are tested.

---

> ### Author Rebuttal · Authors · 2024-08-17
>
> We appreciate your constructive comments for improving the paper, and please find our responses below.
>
> __1.I would appreciated a more detailed discussion (challenges and opportunities) on the distinct behavior of different geographic regions and how insights from one region can be leveraged in other regions (aka transfer learning). The authors touch upon this point just before the final conclusion section, but it would be interesting if they could put some hypotheses.__
>
> Thanks for the suggestion. Differences do exist across geographic regions, particularly concerning surface reflectance, which affects the diffuse and reflected radiation, and weather patterns, which influence cloud movement. These regional variations present both challenges and opportunities for transfer learning. By leveraging general cloud movement information alongside region-specific patterns, transfer learning can enhance the accuracy and generalization of solar forecasting models. Hypothetically, one could investigate whether specific regional features, such as prevalent weather systems or surface albedo characteristics, could be incorporated into the transfer learning process to help improve model performance when applied to new geographic areas. In addition, knowledge-guided machine learning may be a promising direction for the transfer, as the known relationships among variables can be incorporated during the training process to regularize the model and reduce overfitting. This may help enhance the transferability of the model. We expect this to be particularly useful in challenging scenarios when there is a large distinction in the climate conditions of the source and target regions.
>
> __2.The problem formulation can be improved and described more clearly. What are the input features and what are considered ground-truth labels in this ML-modeling set-up?__
>
> We thank the reviewer for pointing this out. For area-based forecasting, both input and labels are satellite-based solar radiation images. For point-based, the inputs are ground-measured solar radiation, three satellite bands, and SZA and the label is ground-measured solar radiation.  We will revise the task formation to make it clearer.
>
> __3.Persistence model appeared ad-hoc, introduce it better, especially for people not familiar with the field__
>
> The persistence model is commonly adopted in previous solar forecasting tasks, especially short-term forecasting [1]. It has a lower computational cost and sometimes shows higher accuracy than complex models [2]. This technique adopts the concept of climate a day ahead and is expected to remain similar to the day before [3]. The short-term persistence model assumes a constant clearness index, representing unchanged atmospheric conditions, within the forecasting horizons. For long-term forecasting, a long-term persistence model assumes unchanged atmospheric conditions from the previous day at the same time. We will introduce it better in the corresponding sections.
>
> [1]https://doi.org/10.1016/j.rser.2020.109792
>
> [2]https://doi.org/10.1016/j.esd.2022.02.002
>
> [3]https://doi.org/10.1016/j.egypro.2017.12.736
>
> __4.Can you introduce MBD in the text (maybe also RMSE for consistency). In Figure 2, the title of the middle graph is rBias. Should this be rMBD?__
>
> We will include an introduction to both Mean Bias Deviation (MBD) and Root Mean Square Error (RMSE) in the relevant section to enhance clarity
>
> __5.Table 1: Please indicate that values are in percent (%).__
>
> We will add "(%)" to the table header in Table 1.
>
> __6.One thing I don’t quite understand is why the video-swin shows the lowest bias by far in the experiment under different land cover types (Fig. 2), but doesn’t show generally low bias under various cloud conditions in Table 1. Can you discuss on this point?__
>
> We sincerely thank the reviewer for highlighting this issue. Upon a thorough review of the code and results, we identified a misalignment when processing land cover data causing the land cover types not matched to the predicted results. We have re-aligned this, and the updated results are provided in the pdf, which correspond well with those in Table 1. Additionally, we carried out another round of reviews on the entire analysis code and ensured that this was the only misalignment. We greatly appreciate your meticulous examination in helping identify this.
>
> __7.In Line 304, you describe “The LSTM-a model shows lower FS for the initial time steps but demonstrates a more pronounced increase in FS as the time step advances.” I don’t conclude the same when looking at the graph, in the very early time steps, the other models have similarly low FS value. I would rather conclude that the FS score is more variable compared to the curves of the other models?__
>
> We agree that your description is more accurate. We will update it to more variable curves than other models.
>
> __8. At the time of submission, the datasets are provided through a google drive link. However, the authors allocated already a Zenodo repo with DOI where they will release the dataset under an open license before publication.__
>
> We will make sure to make the dataset on Zenodo publicly visible before the publication of the paper.
>
> __9.The code is available on a github repo. The license of the codes is missing, both on the github repo and in the Supp. Material.__
>
> We thank the reviewer for pointing this out. We have added the license to the code (MIT License).
>
> __10.Abstract: Detail short- and long-term__
>
> We will define short and long-term forecasting in the abstract.
>
> __11.Line 199: Typo: insufficient -> sufficient__
>
> We will revise "insufficient" to "sufficient" in the manuscript.
>
> __12.Supp. Material: Empty C Additional results__
>
> Thanks for pointing this out. We forgot to delete the title earlier and have now removed it.

---

> > ### Comment · Reviewer_aSBK · 2024-08-30
> > **Rebuttal response**
> >
> > I would like to thank the authors for taking my feedback constructively and addressing all my concerns in the above rebuttal. I hope that the authors will incorporate their responses into the camera-ready version of the paper as promised, at which point I remain in favour of its acceptance for publication.

---

### Official Review · Reviewer_UCVf · 2024-08-03
**Well written and comprehensive dataset for solar forecasting**

**Rating:** 7
**Confidence:** 3

**Review:**

The paper is clearly written, with a good structure that outlines the specific problem, existing work and limitations, and how the authors create the new dataset. It was good to see that the dataset submitted contained not just GOES and other satellite data, but other data that is relevant for solar forecasting (physics based radiation, in-situ data, vegetation data).
The experiments suggested are reasonable representations of the real-world problem, along with good baselines proposed and developed.

**Strengths:**

1. Good collection of data related to solar forecasting, homogenizing various sources of different spatial and temporal resolutions
2. Good coverage of locations around the world, in varying terrain
3. Reasonable task definitions (predicting solar irradiance at broad areas, and specific locations)
4. Reasonable baseline models, and metrics for evaluating the quality of solar forecasting models

**Additional Feedback:**

The SEVIR dataset is mentioned as a reference for ML-ready datasets, which is a good template to follow. The same could be done for near-/short solar forecasting using the data suggested in this paper. The code provides for a way to construct different kinds of datasets, but having a SEVIR-style ready-to-go dataset would be helpful for other researchers to use a standardized task.

As mentioned earlier, it would also be good to have data from more years after 2018.

**Clarity:**

The paper is well written, with a good structure that clearly outlines the problem statement, dataset construction, and experiments.

**Correctness:**

The dataset is constructed in a reasonable way, with metrics and experiments that are relevant to the real-world task of predicting solar irradiance in the near future.

**Documentation:**

The Github link is well structured and contains sufficient details to reproduce the models, however, the Zenodo dataset is not visible publicly.

**Ethics:**

No ethics concerns suspected.

**Limitations:**

The paper correctly mentions the fairness impact of solar forecasting across geographies, which is a known concern since meteorological data is not collected uniformly across the world.
One way to examine the effect of biased data collection on models is to not hold out locations randomly, but hold out locations in specific regions entirely (e.g., ignore all locations in Australia to see if models generalize well there). This could be an additional benchmark task (e.g., zero-shot or one-shot learning to test generalizability of models to new regions).

Data is also from 2018, which makes it difficult to test model generalization across time (to same seasons in future years).

**Opportunities For Improvement:**

Data from NWPs are often used as inputs for predicting solar irradiance, and would be useful to include here as well. E.g., solar flux and cloud cover predictions from global models like IFS/GFS or reanalysis like ERA5, or from regional models like HRRR would be used in real-world prediction tasks, and would be valuable in this dataset as well. This becomes more relevant for long-term forecasting.

**Relation To Prior Work:**

Yes, the paper describes related and prior literature.

**Summary And Contributions:**

This submission provides a dataset to train and evaluate models for solar irradiance nowcasting. The submission includes relevant data from satellite feeds, as well as outputs of physical models and in-situ observations, and auxiliary data that impacts solar power generation. The data and corresponding models are quite relevant to the task of short-range solar power forecasting.
While a good submission, data from a longer temporal period would be valuable to allow evaluation of models generalizing over time ("does a model trained on 2018 data perform well in 2020?").

---

> ### Author Rebuttal · Authors · 2024-08-17
>
> We appreciate your constructive comments for improving the paper, and please find our responses below.
>
> __1. Data from NWPs are often used as inputs for predicting solar irradiance, and would be useful to include here as well. E.g., solar flux and cloud cover predictions from global models like IFS/GFS or reanalysis like ERA5, or from regional models like HRRR would be used in real-world prediction tasks, and would be valuable in this dataset as well. This becomes more relevant for long-term forecasting.__
>
> We thank the reviewer for this suggestion. We agree that incorporating NWP data, particularly meteorological variables, is important for long-term forecasting. We are working on including these data. Once it is completed, we will add it to the Zenodo page and create a new version ID of the dataset to reflect the addition. We have also added this plan to the future work section.
>
> __2. The paper correctly mentions the fairness impact of solar forecasting across geographies, which is a known concern since meteorological data is not collected uniformly across the world. One way to examine the effect of biased data collection on models is to not hold out locations randomly, but hold out locations in specific regions entirely (e.g., ignore all locations in Australia to see if models generalize well there). This could be an additional benchmark task (e.g., zero-shot or one-shot learning to test generalizability of models to new regions).__
>
> We completely agree that holding out locations in specific regions can better assess the effect of biased data on the forecasting models and could be a valuable additional benchmark task. Our data format is also well-suited to support such tasks. We have incorporated this suggestion by adding additional discussion about the task in Section 5 to highlight its potential for evaluating model generalizability.
>
> __3. Data is also from 2018, which makes it difficult to test model generalization across time (to same seasons in future years).__
>
> We thank the reviewer for this suggestion. We agree that including data from other years would enhance the ability to test model generalization and contribute to future fairness studies, which is also in our plan as mentioned in the future work. In response, we will add data over seven SURFRAD areas for the year 2019 in the new version (together with the addition based on the first suggestion). We will continue to expand the temporal coverage of the dataset through future developments.
>
> __4. The Github link is well structured and contains sufficient details to reproduce the models, however, the Zenodo dataset is not visible publicly.__
>
> We will release and make the data publicly visible on Zenodo upon acceptance of the paper. Currently for review the dataset is provided through a Google Drive link. All the additions to the data will continue to be publicly available.

---

### Decision · Program_Chairs · 2024-09-26

**Decision:**

Accept (Poster)

**Comment:**

The reviewers see several strengths in the paper:
- s1. important application: solar forecasting.
- s2. data integration of several sources: satellites and ground measurements.
- s3. clear task description and experimental protocol.
- s4. plausible selection of models in the benchmark.

But they also discussed several weaknesses:
- w1. data from numerical weather prediction not included.
- w2. some unplausible findings:
     i) performance of video-swin under different land cover types
	   vs. different cloud conditions.
     ii) better performance on hard examples than on easy examples for
	    some metrics.
- w3. benchmark does not show big differences between different models.

In their rebuttal, the authors explained w2 "unplausible findings"
well and also discussed the other points.

Overall I recommend to accept the paper.